# Chromatin in 3D distinguishes dMes-4/NSD and Hypb/dSet2 in protecting genes from H3K27me3 silencing

David Depierre[1,*], Charlène Perrois[1,*], Naomi Schickele[1], Priscillia Lhoumaud[1], Mahdia Abdi-Galab[1], Olivier Fosseprez[1] (ORCID), Alexandre Heurteau[1], Raphaël Margueron[2], Olivier Cuvier[1] (ORCID)

**Cell type-specific barcoding of genomes requires the establishment of hundreds of heterochromatin domains where heterochromatin-associated repressive complexes hinder chromatin accessibility thereby silencing genes. At heterochromatin–euchromatin borders, regulation of accessibility not only depends on the delimitation of heterochromatin but may also involve interplays with nearby genes and their transcriptional activity, or alternatively on histone modifiers, chromatin barrier insulators, and more global demarcation of chromosomes into 3D compartmentalized domains and topological-associating domain (TADs). Here, we show that depletion of H3K36 di- or tri-methyl histone methyltransferases dMes-4/NSD or Hypb/dSet2 induces reproducible increasing levels of H3K27me3 at heterochromatin borders including in nearby promoters, thereby repressing hundreds of genes. Furthermore, dMes-4/NSD influences genes demarcated by insulators and TAD borders, within chromatin hubs, unlike transcription-coupled action of Hypb/dSet2 that protects genes independently of TADs. Insulator mutants recapitulate the increase of H3K27me3 upon dMes-4/NSD depletion unlike Hypb/dSet2. Hi-C data demonstrate how dMes-4/NSD blocks propagation of long-range interactions onto active regions. Our data highlight distinct mechanisms protecting genes from H3K27me3 silencing, highlighting a direct influence of H3K36me on repressive TADs.**

## Introduction

Eukaryotic genomes are globally partitioned into two major active and inactive A and B compartments, or euchromatin and heterochromatin, respectively, which largely contribute to cell-type gene expression programming. High-resolution genome mapping of the major histones post-translational modifications and of their chromatin-associated proteins writing, reading, or erasing these histone marks (Filion et al, 2010; Kharchenko et al, 2011; Ho et al, 2014; Evans et al, 2016) revealed that such partitioning actually involves more than two distinct chromatin states. As a result, various combinations of factors and marks give raise to chromatin states more or less permissive to transcription, which largely depends on chromatin accessibility (Carelli et al, 2017). Transcription may further impede on 3D organization of chromatin by favoring interactions among compartmentalized domains (Rowley et al, 2017). To what extent transcriptional activity and the partitioning of chromatin into distinct domains may mutually influence each other remains unclear.

Cell type-specific silencing of gene expression requires notably polycomb repressive complexes (PRC1 and 2) that contribute to mediate repression by di/trimethylation of lysine 27 on histone H3 (H3K27me2/3) and binding of CBX and EED subunits to these marks (Margueron et al, 2009). H3K27me2/3 distributes into tenths of kilobases long facultative heterochromatin domains defining relatively inaccessible chromatin. The resulting organization into hundreds of successive heterochromatin and euchromatin blocks generates a barcoding of the genome that is specific to each cell-type. The EZH subunit of PRC2 further tri-methylates H3K27 over neighbor nucleosomes (Holoch & Margueron, 2017; Poepsel et al, 2018; Laugesen et al, 2019), leading to polycomb-mediated silencing through the spreading of the H3K27me3 repressive mark (Talbert & Henikoff, 2006), once PRC2 has been loaded onto discrete nucleation sites (Oksuz et al, 2018). Alternative establishment of domains can occur in absence of H3K27me3, involving Suz12-mediated PRC2 loading (Højfeldt et al, 2018). It remains not totally clear how active genes are protected from the spreading of H3K27me3 domains that must be tightly regulated to maintain cell-type specific barcoding of the genome.

Of interest, the trimethylation of lysine 36 on histone H3 (H3K36me3) occurs co-transcriptionally (Krogan et al, 2003; Lhoumaud et al, 2014). Although H3K36me3 can antagonize H3K27me3 propagation by inhibiting PRC2 activity (Schmitges et al, 2011; Yuan et al, 2011; Voigt et al, 2012), it remains unclear if H3K36me3 blocks H3K27me3 independently of transcription. Also, the two PRC2 subunits PHF1 and PHF19 bind H3K36me3 (Ballaré

[1]Chromatin Dynamics and Cell Proliferation, Center of Integrative Biology, Molecular, Cellular and Developmental Biology (MCD/UMR5087), CNRS, Université Paul Sabatier de Toulouse, Toulouse, France   [2]Institut Curie, Paris Sciences et Lettres Research University; INSERM U934/ CNRS UMR3215, Paris, France

Correspondence: olivier.cuvier@univ-tlse3.fr
*David Depierre and Charlène Perrois contributed equally to this work

et al, 2012; Musselman et al, 2012; Finogenova et al, 2020). In *Caenorhabditis elegans*, the histone methyltransferase (HMT) MES-4 ensures both di- and tri-methylation of H3K36 (H3K36-me1/me2/me3) that may antagonize H3K27me3 (Gaydos et al, 2012; Evans et al, 2016; Ahringer & Gasser, 2018) and MET-1 tri-methylates H3K36 (H3K36me3) (Cabianca et al, 2019). Other eukaryotes that possess facultative heterochromatin, also contain two or more HMTs, including in *Drosophila* with dMes-4 (NSD1/2/3 homolog) and HypB/dSet2 (SETD2 homolog), which may be responsible of transcription-coupled H3K36-me2 and -me3, respectively. It remains to be shown whether it is the di- or tri- methylated state of H3K36 that can block H3K27me3 self-propagation (Huang & Zhu, 2018; Streubel et al, 2018). dMes-4/NSD is recruited to chromatin by insulator-binding proteins like CTCF and Beaf-32, thereby regulating genes (Lhoumaud et al, 2014). Furthermore, tri-methylation (H3K36me3) by Hypb/dSet2 occurs when this HMT interacts with the C-terminal domain of the elongating form of RNA polymerase II (Pol II), upon phosphorylation by PTEF-b/Cdk9 (Krogan et al, 2003; Kizer et al, 2005; Edmunds et al, 2008; Wagner & Carpenter, 2012). This triggers Pol II release from pausing along with the recruitment of HypB/dSet2, that is, coupled with elongation (Venkatesh & Workman, 2013; Lhoumaud et al, 2014). Yet it is unclear which of transcription or H3K36me per se may be necessary for the demarcation of active domains from repressed H3K27-methylated domains.

Here, we performed a genome-wide comparative analysis of the H3K27me3 histone marks in control versus dMes-4/NSD- or Hypb/dSet2-depleted cells. The depletion of Hypb/dSet2 leads to some H3K27me3 spreading, notably over genes flanking a H3K27me3 repressive domain border that do not coincide with a topological-associating domain (TAD) border. In contrast, dMes-4/NSD protects genes flanking a TAD border and that can assemble into 3D chromatin hubs. Accordingly, our novel Hi-C data show that depletion of dMes-4/NSD extend long-range interactions out of the inactive TADs, to the same regions where H3K27me3 spreading is also detected, unlike for Hypb/dSet2. Our results highlight how dMes-4/NSD and Hypb/dSet2 may help sustaining the active state of genes depending on the positioning of the heterochromatin–euchromatin borders at the border of a TAD, or not.

# Results

## Both dMes-4/NSD and HypB/dSet2 protect genes from spreading of H3K27me3

We sought to test which of H3K36me2- or H3K36me3-methylated states may antagonize the spreading of H3K27me3. Both H3K36-me2 and -me3 mark euchromatin domains flanking repressive heterochromatin TADs (Fig 1A). The average distribution of H3K36me marks over hundreds of borders separating euchromatin from H3K27me3 heterochromatin, highlighting a local increase in H3K36me2 at borders (Figs 1B and S1A), in line with the interaction of dMes-4/NSD with chromatin insulators (Lhoumaud et al, 2014). In contrast, H3K36me3 demarcated more globally euchromatic from heterochromatin domains in S2 cells, possibly reflecting the interaction of Hypb/dSet2 with Pol II elongating over genes (Huang

& Zhu, 2018; Streubel et al, 2018). Quantitative analysis of the levels of H3K36 and H3K27 methylations indicated that both H3K36-me2 and -me3 globally anti-correlated with H3K27me3 levels as confirmed by analyzing the signal at scales of individual loci by scatter plot (Fig 1C) or by heatmaps, ranking loci according to levels of H3K36me2, H3K36me3, or H3K27me3 (Figs 1D and E and S1B and C). Such rankings showed a mutual exclusion of H3K36me2/3 with H3K27me3, with few regions harboring both types of histone methylation, supporting a global antagonism between H3K36-me2 or -me3 and H3K27me3, at genome-wide scales (Fig S1D).

To address more directly how H3K36 methylation regulates H3K27me3, we next sought to impair either di- or tri-methylation of H3K36 (Fig 2A). Also, many multicellular eukaryotes possess at least two major HMTs specific for H3K36-me2 or -me3 (Huang & Zhu, 2018). In particular, the two major HMTs of H3K36 in *Drosophila melanogaster*, namely, dMes-4/NSD (NSD homolog) and HypB/dSet2 (SETD2 homolog), are responsible for H3K36-me2 and -me3 deposition, respectively. Accordingly, depletion of dMes-4/NSD or of Hypb/dSet2 (validated in Fig 2B) mostly reduced chromatin-associated H3K36-me2 and -me3 levels compared with control levels, respectively (Fig 2C). Of note, depletion of Hypb/dSet2 did not lead to a loss of H3K36me2 levels yet it strongly impaired H3K36me3 compared with control histone H3, whereas dMes-4/NSD impaired H3K36me2 and a slight decrease of H3K36me3 (Fig S2A–C). As such, this system could readily test if depleting Hypb/dSet2 may influence H3K27me3 through H3K36me3, independently of changes in H3K36me2 levels. In the case of dMes-4/NSD, although it led to a net decrease in H3K36me2, one may not totally exclude the possibility that it can influence H3K27me3 due to a slight decrease in H3K36me3 levels.

Depletion of dMes-4/NSD led to significantly increased H3K27me3 levels at hundreds of genomic sites, as illustrated (Fig 2D). Such increases in H3K27me3 levels supported a role of dMes-4/NSD as a cofactor-regulating chromatin accessibility (Lhoumaud et al, 2014). The depletion of Hypb/dSet2 similarly led to increased H3K27me3 levels at hundreds of sites. The increases in H3K27me3 levels observed upon both depletions were significant and reproducible, as shown for depletion of dMes-4/NSD and Hypb/dSet2 (Fig S2D and E), thus validating that both HMTs generally antagonizes H3K27me3.

Genomic bins harboring increased H3K27me3 levels upon depletion of either HMT were preferentially encountered in gene bodies compared with random distribution (Fig 2E; dMes-4/NSD and Hypb/dSet2; 61% and 59%; *P*-values of $1 \times 10^{-126}$ and $1 \times 10^{-17}$, respectively; see the Materials and Methods section), and more rarely in intergenic regions. Accordingly, heatmaps showed that upon both depletions, the net increase in H3K27me3 levels most often mapped over gene bodies (Fig 2F). In the case of HypB/dSet2, this HMT was shown to interact with Pol II elongating along gene bodies (Li et al, 2003; Albert et al, 2014; Huang & Zhu, 2018; Streubel et al, 2018), which could possibly account for why it counteracted H3K27me3 over gene bodies. Supporting this view, the influence of these HMTs was only detected in the presence of a gene at H3K27me3 borders (Fig S2F and G). Furthermore, ranking genes according to their increase in H3K27me3 levels upon dMes-4/NSD KD showed a tendency to sort out genes also harboring increasing H3K27me3 upon Hypb/dSet2 KD (Fig 2G; *P*-value of $2.5 \times 10^{-5}$) (see the Materials and Methods section). Taken altogether, our data thus indicated that compromising H3K36me2 or H3K36me3 levels by

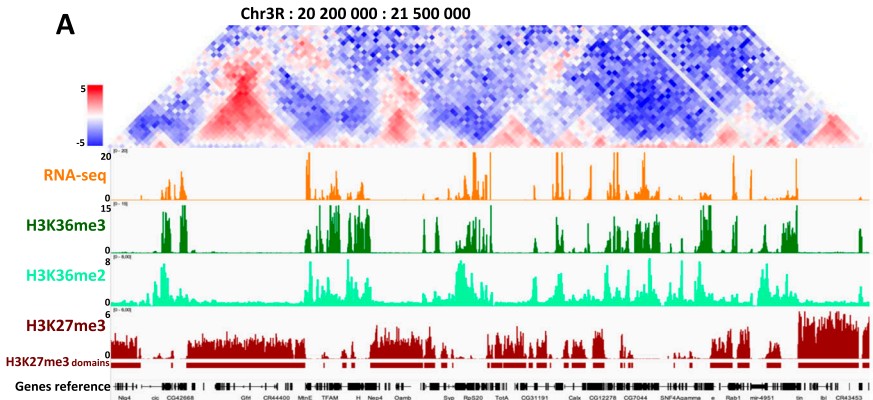

**A**

Chr3R : 20 200 000 : 21 500 000

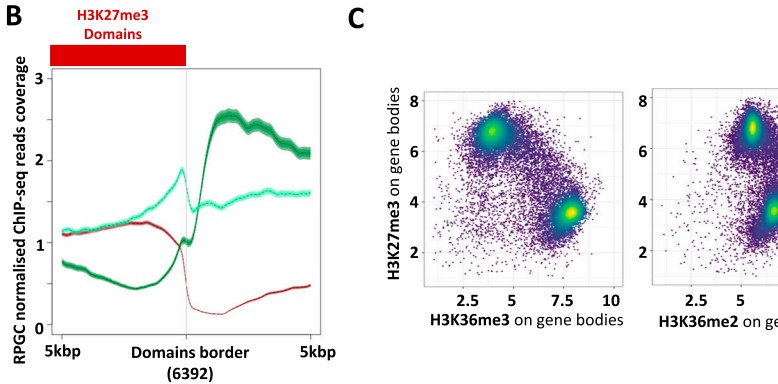

**B**

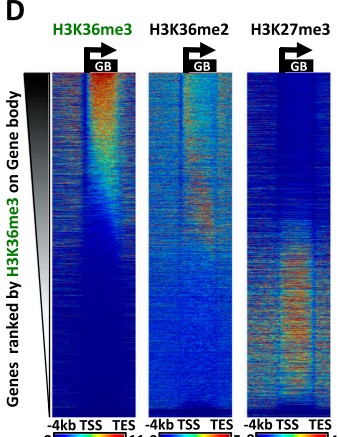

**C**

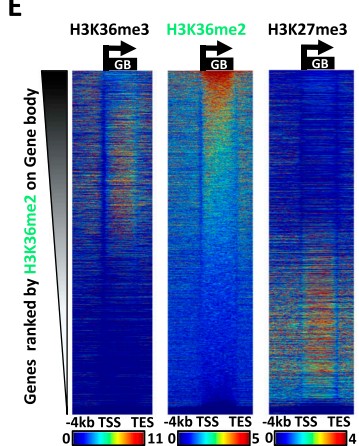

**Figure 1.  Genomic distribution of H3K36 and H3K27 methylations fits with topological and functional domains.**

**(A)** Visualization of Hi-C contact frequency matrix, RNA-seq reads, and ChIP-Seq reads of (from top to bottom) H3K36me3, H3K36me2, and H3K27me3 on the indicated region of chromosome 3. Reads were binned and smoothed at 10 bp scales and RPGC normalized. The red stripes indicate H3K27me3 domains as detected with the normR R package (H3K27me3 relative to input, FDR < 0.0001) for domains > 1,500 bp, for a total of 3,196 domains detected (see the Materials and Methods section). The last track shows referenced genes (in black). **(B)** Average profiles of H3K36me3, H3K36me2, and H3K27me3 around the H3K27me3 6392 (3,196 × 2) domain borders (oriented with H3K27me3 domains systematically on the left). H3K27me3 domains were computed with normR applied to the corresponding ChIP-seq data (this study; see the Materials and Methods section). Bold line represents averaged signal and faded color represents the confidence interval 95%. **(C)** Scatter plot showing quantification of the levels of H3K27me3 and of H3K36me3 (left panel) or of H3K27me3 and of H3K36me2 (right panel) quantified on all gene bodies. Gradient colors (from purple to yellow) represent the density of genes. **(D)** Heatmaps showing the ChIP-Seq of (from left to right) H3K36me3, H3K36me2, or H3K27me3 all ranked by the levels of H3K36me3 reads on gene bodies (see the Materials and Methods section). The heatmaps show all 17,453 genes for a window spanning −4 kbp upstream of transcription start sites down to 2 kbp downstream of transcription end sites (with scaled genes in-between; see the Materials and Methods section). **(E)** Same as (D) after ranking heatmaps with H3K36me2.

**D**

**E**

either dMes-4/NSD or Hypb/dSet2 depletion could both impair the blocking of H3K27me3 spreading, as notably detected over gene bodies.

### Depletion of dMes-4/NSD or Hypb/dSet2 impairs the expression of hundreds of genes

The observed variations in H3K27me3 upon dMes-4/NSD or Hypb/dSet2 depletion might not induce a robust change in the positioning of core H3K27me3 domains. Rather, a relative increase of this mark may occur, prompting us to test whether it was functionally relevant, for example, if sufficient to detect and influence on gene expression. Gene expression was globally quantified by RNA-seq followed by differential expression analysis using limma (Ritchie et al, 2015) (see the Materials and Methods section). Depletion of dMes-4/NSD led to 260 and 224 genes to be, respectively, down- and up-regulated compared with control. Similarly, depletion of Hypb/dSet2 led 534 and 381 genes to become down- and up-regulated, respectively (Fig 3A). Hierarchical clustering on gene expression identified distinct classes of positively or

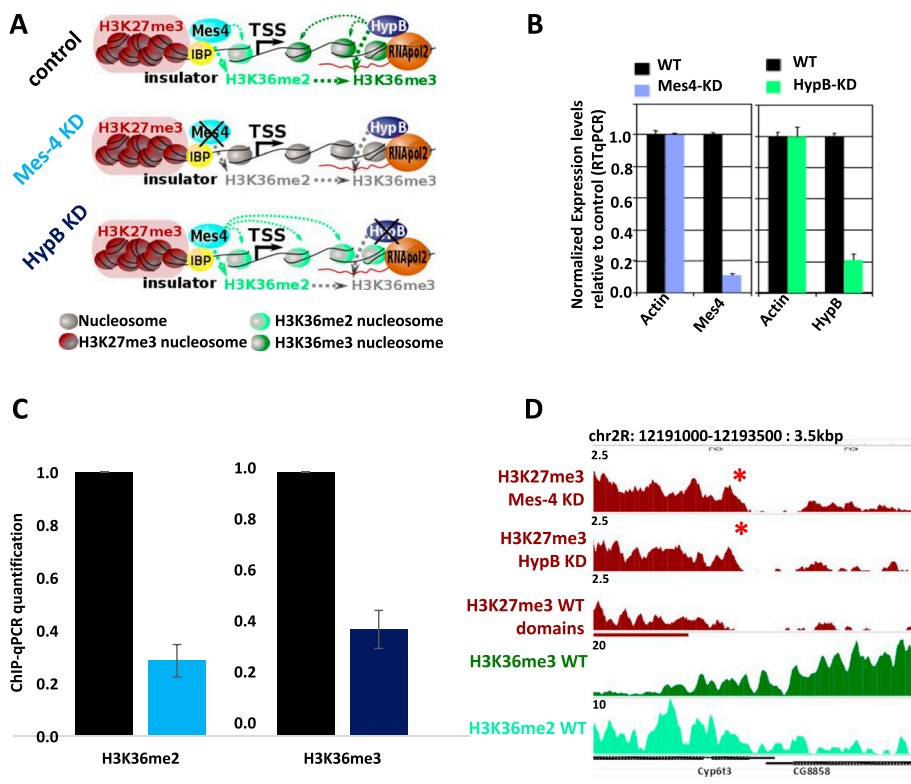

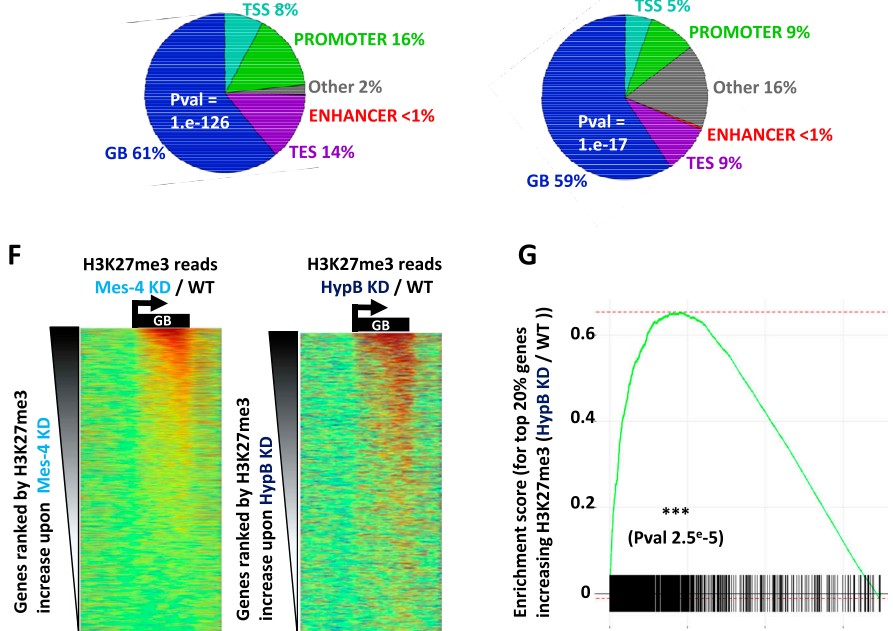

**Figure 2.  Both H3K36 methyltransferases dMes-4/ NSD and HypB/dSet2 counteract H3K27me3 spreading.**

**(A)** Scheme of the experimental strategy by siRNA-mediated depletion of dMes-4/NSD (the histone methyltransferase for H3K36me2) or of HypB/dSet2 (the histone methyltransferase required for H3K36me3). The top scheme represents the WT condition, the middle the situation upon depletion of dMes-4/NSD, resulting in the reduction of H3K36me2 (dashed grey arrows), and the lowest scheme the situation upon HypB/dSet2 depletion that results in a reduction of H3K36me3 (whereas H3K36 dimethylation may remain unchanged). Dashed arrows represent methyltransferase activity (in grey when disabled). **(B)** RT-qPCR assessing the depletion of dMes-4/NSD and HypB/dSet2 compared with control genes (see the Materials and Methods section). **(C)** ChIP-qPCR quantifications showing loss of H3K36me2 upon depletion of dMes-4/NSD and loss of H3K36me3 upon HypB/dSet2 depletion on 11 gene bodies (see Table S2 for primers). **(D)** Genomic browser showing significant increases in H3K27me3 around Cyp6t3 gene under both conditions of HypB/dSet2 and dMes-4/NSD depletion. Signal is shown as reads coverage RPGC normalized (values indicated in Y axis scale) on the chr2R from 12,190,000 to 12,193,500 bp. **(E)** Pie chart showing distribution of bins harboring increasing H3K27me3 levels upon depletion of HypB/dSet2 or dMes-4/NSD over genomic features (TSS, transcription start sites; TES, transcription end sites; GB, gene bodies). Bins with increase were detected and significantly validated with an FDR of 10–4 using enrichR function of the normR package. *P*-values on pie chart were calculated using a Fisher exact test comparing bins with increasing H3K27me3 to randomly distributed bins and validate enrichment of bins with increases on gene bodies. **(F)** Differential heatmap showing the net variations in H3K27me3 ChIP-Seq reads upon depletion of dMes-4/NSD (left, 5,298 genes) or HypB/dSet2 (right, 1,924 genes) compared with WT control. Genes were ranked by H3K27me3 norm. diff. score (see the Materials and Methods section). The heatmap window spans –4 kbp upstream of TSSs down to 2 kbp downstream of TESs with scaled gene bodies in the intervening window. **(G)** Genes set enrichment analysis testing whether ranking of genes according to their increases in H3K27me3 levels upon dMes-4/NSD depletion (compared with control) can predict their susceptibility to be exposed to increasing H3K27me3 levels upon depletion of Hypb/dSet2 (top 20% of genes: 1,300 genes; black stripes) or not (complementary list of 80% of control genes: 5,200 genes; white stripes). The enrichment analysis tests whether the variations in H3K27me3 upon dMes-4/NSD-KD can selectively sort out the genes harboring most significant H3K27me3 spreading (top 20%) upon Hypb/dSet2 KD (*P*-value of 2.5 × 10$^{-5}$; see the Materials and Methods section).

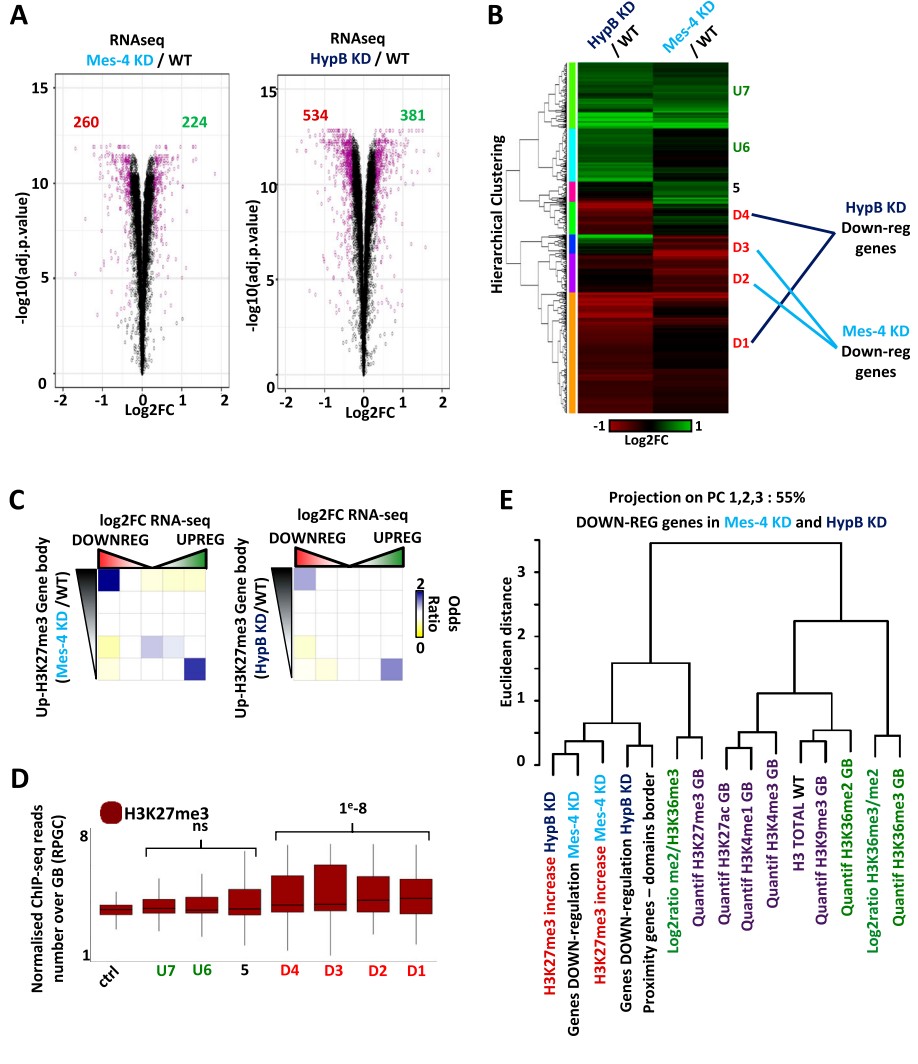

**Life Science Alliance**

**Figure 3. dMes-4/NSD and HypB/dSet2 protect genes from being silenced by heterochromatinization.**

**(A)** Volcano plot showing the $\log_2$ fold change of differentially expressed or control genes (in purple or black, respectively) as quantified by RNA-seq upon dMes-4/NSD or HypB/dSet2 depletion (left and right plots, respectively) compared with control cells (see the Materials and Methods section). **(B)** Hierarchical ascendant clustering analysis among differentially expressed genes identifies two down-regulated clusters of genes upon dMes-4/NSD or HypB/dSet2 depletion (D2 and D3; D1 and D4, respectively) and three up-regulated clusters (U5, U6, and U7). **(C)** Intersection matrix between quintile of genes with increasing H3K27me3 and differentially expressed genes upon dMes-4/NSD or HypB/dSet2 depletion (left and right, respectively). Odds ratio Fisher exact test are represented from blue to yellow for enrichment to depletion of the intersected gene quintile. Note that only the first quintile of genes (i.e., 20% of active genes, ~1,500 genes) with H3K27me3 increases is enriched in down-regulated genes quintile. **(D)** Boxplot quantifying the levels of H3K27me3 in WT condition for differentially expressed gene clusters. Significant differences are validated comparing pooled up- or down-regulated genes to control genes using a Wilcoxon test. **(E)** Dendrogram representing the projected Euclidean distances as calculated by principal component analysis; three firsts PCs explain 55% of the signal variability as shown in Fig S3E; (see the Materials and Methods section). This representation summarizes correlations on the three first PCs between quantifications, differential scores, and metrics used in this study. The more two variables are close in the dendrogram, the more they are projected on the same principal(s) component(s), and thus correlated. Note that down-regulated genes correlate with genes harboring increasing H3K27me3 levels upon depletion of HypB/dSet2, for genes in proximity to heterochromatin domain borders. NB: proximity genes–domain borders are defined as the distance in base pair between a gene and the closest H3K27me3 domain border.

negatively expressed genes specifically encountered in one of these KMT depletions (Figs 3B and S3A; see the Materials and Methods section). Therefore, dMes-4/NSD and Hypb/dSet2 may often regulate distinct genes. Importantly, genes harboring increasing levels of H3K27me3 levels were specifically enriched in down- yet not in up-regulated genes, as shown upon either depletion of dMes-4/NSD and Hypb/dSet2 (Figs 3C and S3B and C). Moreover, the genes from down-regulated clusters, tended to be moderately expressed compared with up-regulated one's (Fig S3A) they localized in regions with higher H3K27me3 levels (Fig 3D; $P$-value < $1 \times 10^{-8}$ and 1, respectively). Thus, although the variations in H3K27me3 may not reflect a change in the positioning of core H3K27me3 domains, depletion of dMes-4/NSD and Hypb/dSet2 could still expose genes to silencing through spreading of H3K27me3 marks. Such phenomenon is mostly encountered within gene bodies and not in enhancers localized outside of gene bodies (Fig S3D). A global analysis by principal component analysis taking the first 3 PC's counting for 55% of variability explanation (Figs 3E and S3F) confirmed that gene expression down-regulation was tightly correlated to increases in H3K27me3.

Moreover, computing gene distances to H3K27me3 borders showed that down-regulation was linked to the proximity of genes to such borders (Fig 3E), which highly correlated with increasing levels of H3K27me3, which was not observed for up-regulated genes (Fig S3E). As such, these results confirmed that in absence of dMes-4/NSD or Hypb/dSet2, H3K27me3 spreading occur over genes near H3K27me3, specifically exposing them to down-regulation. Of note, gene ontology analysis showed that the down-regulated genes were enriched in genes regulating developmental functions particularly for Hypb/dSet2 (Table S1) and less for dMes-4/NSD. Our data thus highlighted a global link between a positive function in gene expression of dMes-4/NSD and Hypb/dSet2, the latter being enriched in moderately expressed, developmentally regulated genes that may be exposed to polycomb-mediated repression. Of note, quantification of RNA-seq reads on PRC1 and PRC2 subunit showed no changes in expression in dMes-4/NSD KD or Hypb/dSet2 KD, indicating that increases in H3K27me3 in those depletions were not an indirect consequence of polycomb-associated genes expression variations (Fig S3G).

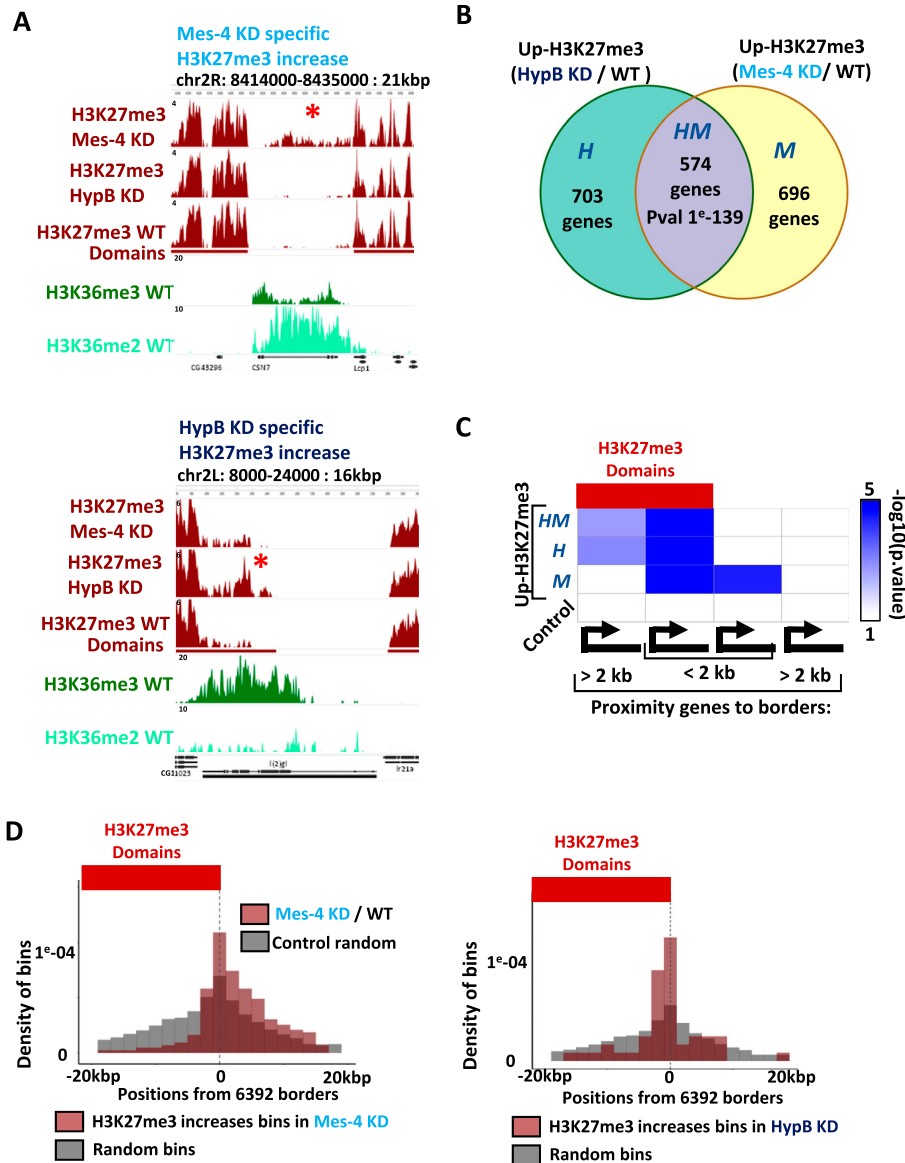

**Figure 4. dMes-4/NSD and HypB/dSet2 counteract H3K27me3 spreading on distinct heterochromatin borders.**

**(A)** Genomic browser showing examples of regions/genes with variations in H3K27me3 unique to dMes-4/NSD (upper panel) or HypB/dSet2 (lower panel) depletions. **(B)** Venn diagram comparing the overlap between the lists of genes with highest H3K27me3 increases upon depletion of dMes-4/NSD and of HypB/dSet2 (top 20% of genes ranked according to their differential H3K27me3 score, chosen accordingly to the first quintile of H3K27me3 increase genes enriched in down-regulated genes in Fig 3C) (see the Materials and Methods section). Genes were classified depending on the observed increasing H3K27me3 levels upon depletion of either HypB/dSet2 ("H") or dMes-4/NSD ("M") or upon both depletions (group "HM"; *P*-value of $1 \times 10^{-139}$ as calculated using a Fisher exact test). **(C)** Distribution of sites/genes harboring an increase in H3K27me3 levels upon depletion of dMes-4/NSD (red) or HypB/dSet2 with respect to H3K27me3 borders (position 0) (see the Materials and Methods section). Sites with increasing H3K27me3 were categorized as inside H3K27me3 domain (first column: inside H3K27me3 domains; >2 kbp from a H3K27me3 border) on the H3K27me3 side of the border (second column: inside H3K27me3 domains; <2 kbp from a H3K27me3 border), or outside H3K27me3 yet close to a border (third column: outside H3K27me3; <2 kbp from a H3K27me3 border) or far outside from H3K27me3 domains (fourth column: outside H3K27me3; >2 kbp from a H3K27me3 border). *P*-values were calculated using a Fisher exact test. **(D)** Genomic distribution of bins harboring an increase in H3K27me3 levels upon depletion of HypB/dSet2 or dMes-4/NSD and overlapping with down-regulated genes D1 and D4 for HypB/dSet2 and D2 and D3 for dMes-4/NSD (red) compared with randomly distributed bins with respect to 6,392 heterochromatin borders (position 0, oriented with H3K27me3 systematically on the left). Random bins (in grey) were chosen as a control for both HypB/dSet2 KD and dMes-4/NSD depletions (see the Materials and Methods section).

## Hypb/dSet2 and dMes-4/NSD regulate H3K27me3 levels in distinct chromatin domains

The identification of distinct clusters of genes being down-regulated upon depletion of either dMes-4/NSD or Hypb/dSet2, along with distinct enrichments in specific gene ontologies (Table S1), prompted us to re-evaluate their relative influence on H3K27me3 levels. Although their depletion led to a significant overlap (Fig 4B; >45%; *P*-value of $1 \times 10^{-139}$; Fig S4A and B), many genes harbored higher H3K27me3 levels only upon depletion of dMes-4/NSD ("M"; 696 genes) or of Hypb/dSet2 ("H"; 703 genes). Visual inspection of our ChIP-seq data showed that upon depletion of dMes-4/NSD, genes with increasing H3K27me3 often localized on the euchromatic side of H3K27me3 domain borders (Fig 4A; upper graph), as confirmed by statistical enrichment of the genes uniquely down-regulated by dMes-4/NSD and not Hypb/dSet2 (Figs

4B and S4C), which were enriched on the euchromatic side of borders (Fig 4C). For dMes-4/NSD, 20% of genes showed increased H3K27me3 levels when localizing near a border compared with only 6% or 7% for those localizing in euchromatin or heterochromatin, far from a border (Fig S4D; *P*-value of $1 \times 10^{-40}$ and 1, respectively). This contrasted with genes with higher levels of H3K27me3 uniquely detected upon depletion of Hypb/dSet2 (Fig 4B; "H"; 703 genes). This depletion showed a higher influence on genes localized at the heterochromatic side of borders compared with euchromatin (Fig 4B and C; Hypb/dSet2: 17–24% versus 6–8%; *P*-value of $1 \times 10^{-5}$). Overall, sites with both increases of H3K27me3 and down-regulation of genes upon depletion of either dMes-4/NSD or Hypb/dSet2 showed a distinct distribution over the H3K27me3 borders (Fig 4D). Depletion of dMes-4/NSD increased H3K27me3 and impaired expression for genes over the euchromatic side of borders, whereas depletion of Hypb/dSet2 mostly affected genes

on the heterochromatic side of borders. Of note, the levels of expression of the genes regulated by dMes-4/NSD were often higher than those regulated by Hypb/dSet2, as estimated by RNA-seq quantification (Fig S4E). Accordingly, genes regulated by Hypb/dSet2 often localize within H3K27me3 domains or at the inside borders of such domains, where both H3K27me3 and H3K36me3 marks are present (Fig S4F). We thus sought to test whether there is a requirement for transcription that may account for the influence of dMes-4/NSD or Hypb/dSet2 in protecting from increasing H3K27me3 levels, as such increase might be an indirect consequence of gene down-regulation. The increase in H3K27me3 levels was however not dependent on further down-regulation of genes upon depletion of either KMTs, as shown Fig S4G. 701 and 681 genes harbored increasing H3K27me3 levels upon dMes-4/NSD or Hypb/dSet2 depletion, uncoupled from a down-regulation of the genes. Moreover, of the genes regulated by dMes-4/NSD that localize few kb away from a H3K27me3 domain border, it is interesting to note that the increase in H3K27me3 was also detected in the interspace between such borders and the gene (Fig S4H), supporting a role for this KMT in blocking the spreading of H3K27me3 towards euchromatin. Of note, the H3K27me3 domains detected here may possess relatively 1.3x lower H3K27me3 levels as than strongly repressed polycomb domains marking Hox gene clusters (Fig S4I and J). Accordingly, few of the H3K27me3 domains detected here harbor a PRE (Fig S4K), which may explain why the expression of the genes within these domains is not fully repressed. However, genes within such domains harbored significantly (~2x-) higher levels of H3K27me3 as compared with genes outside of such domains (Fig S4J). As such, depletion of dMes-4/NSD or Hypb/dSet2 may exacerbate the repression of genes localized within the H3K27me3 domains identified here, as evidenced by their significant down-regulation.

Genes harboring increasing levels of H3K27me3 upon dMes-4/NSD depletion were often flanking an inactive TAD domain (Fig 5A). Such genes were mostly associated with active compartment Eigen values, as illustrated by the *tsp39D* gene (Fig 5A and B) (see the Materials and Methods section), in contrast to genes with increasing H3K27me3 levels upon Hypb/dSet2 depletion that often localized in inactive TADs, as illustrated for the *crc* gene (Figs 5A and S4F). Accordingly, such genes were associated with lower Eigen values between A and B compartment values (Fig 5B). In agreement, genes protected from increasing H3K27me3 levels by dMes-4/NSD were significantly enriched among active genes harboring higher levels of H3K36me levels as compared with the genes protected by Hypb/dSet2 (Figs 5C and S5A). Of note, Hypb/dSet2 was shown to interact with Pol II (Li et al, 2003; Albert et al, 2014), which may protect active genes from H3K27me3. Supporting this view, transcriptional inhibition was sufficient to render an active gene more dependent on Hypb/dSet2 depletion. Addition of flavopiridol (FP), an inhibitor of Cdk9 activity (Albert et al, 2014), led to efficient inhibition of transcription, as shown (Fig S5B). Providing Hypb/dSet2 had been depleted from cells, such inhibition actually enhanced the increase H3K27me3 levels at active genes (Fig S5C). Our results thus support the view that Hypb/dSet2 may be needed at genes exposed to H3K27me3. Such influence may be exacerbated within the repressive environment of an inactive TAD, as illustrated by *crc* (Fig 5A), that is, when their level of expression is low or blocked.

On the contrary, genes protected by dMes-4/NSD were in euchromatin, harbor higher levels of H3K36me, and unlike for Hypb/dSet2, were enriched in insulator protein-binding sites (Fig 5D; *P*-value < 1 × 10$^{-8}$ and 1, respectively). Accordingly, probing long-range interactions (LRIs) by aggregation of Hi-C data, as previously developed (Liang et al, 2014; Rao et al, 2014) (see the Materials and Methods section), confirmed the tendency for genes protected by dMes-4/NSD to establish long-range contacts (Fig 5E; see below). This tendency may reflect the interactions of genes within 3D clusters, and of note, it distinguishes such genes from those protected by Hypb/dSet2. Taken altogether, our data thus raised the possibility that chromatin hubs and possibly TADs may define a general feature associated with genes protected from H3K27me3 spreading by dMes-4/NSD, unlike for Hypb/dSet2.

H3K27me3 domains represent the mostly marked TADs in Drosophila (Szabo et al, 2019), and about half of inactive TAD boundaries annotated in the study by Ramírez et al (2018) are overlapping with H3K27me3 domain borders identified in this study (Fig S5D). We therefore sought to test whether unlike Hypb/dSet2, dMes-4/NSD might protect genes from heterochromatin depending on such spatial organization into TADs. The ability of dMes-4/NSD or Hypb/dSet2 to protect from H3K27me3 spreading was first tested depending on the co-localization of H3K27me3 borders with a TAD border or not (Figs 5F and S5D and E, left and right plots, respectively). Of note, genes with increasing H3K27me3 levels upon depletion of dMes-4/NSD predominantly distributed over TAD boundaries that overlapped with an H3K27me3 domain borders (Fig 5F; left plot). In contrast, Hypb/dSet2 predominantly regulated genes localizing near an inactive TAD boundary that is not colocalizing with an H3K27me3 domain border (Fig 5F; right plot). Supporting such influences depending on overlaps between H3K27me3 borders and TAD borders, the influence of dMes-4/NSD and Hypb/dSet2 could be validated statistically by directly assessing their influence on averaged H3K27me3 levels as shown (Figs 5G and S5F; *P*-value of 1 and < 1 × 10$^{-16}$ and *P*-value of 1 × 10$^{-8}$ and 1, for dMes-4/NSD and Hypb/dSet2, respectively).

### The dMes-4/NSD demarcates repressive H3K27me3 domains within TADs

Whether histone modifiers and the resulting modifications represent a cause or a consequence of topology has been a recurrent question (Ulianov et al, 2016; Szabo et al, 2019). We sought to test whether the spreading of H3K27me3 might occur independently of any change in 3D organization, or whether this may be coupled to possible changes in topology. To this end, we performed Hi-C in the same cells depleted of dMes-4/NSD or Hypb/dSet2 or control cells. Strikingly, inspection of normalized Hi-C data showed that dMes-4/NSD uniquely led to increase Hi-C counts at the borders of repressive TADs (Fig 6A; see red arrow, Fig S6A), unlike what was observed in control cells or upon depletion of Hypb/dSet2 (Fig S6B and C). Consistently, this phenotype was detected over sites harboring H3K36me levels bordering a repressive H3K27me3 TAD domain (Fig 6A; see ChIP-seq H3K36me and H3K27me3 tracks on top). Hi-C contacts were assessed more systematically on Hi-C profiles aggregated around the sites harboring increasing H3K27me3 levels upon dMes-4/NSD or Hypb/dSet2 depletion. Strikingly, a significant

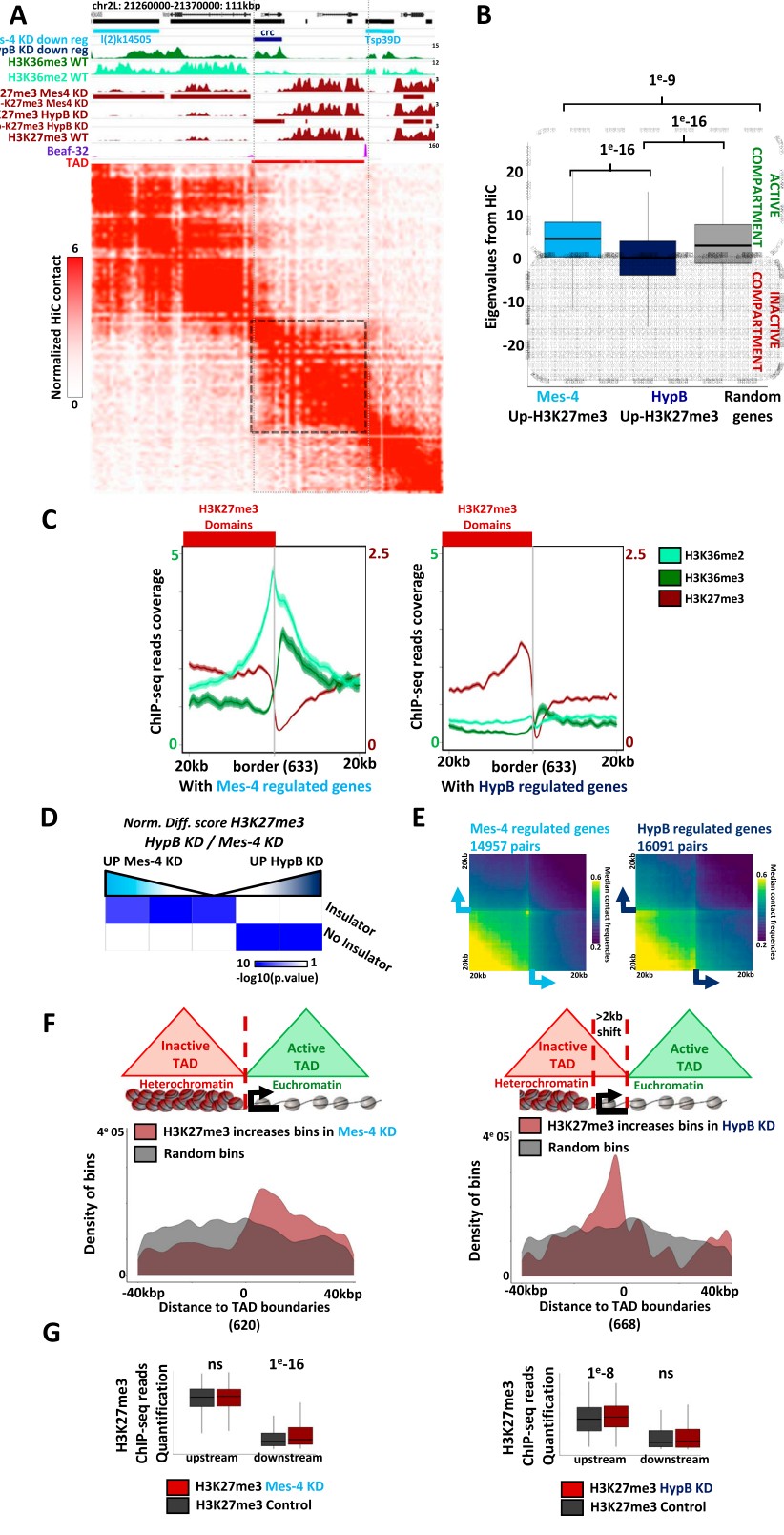

Figure 5. 3D architecture influences dMes-4/NSD or HypB/
dSet2 regulation on border-flanking genes.

**(A)** Hi-C signal visualization on chromosome 2L
(21,260–21,350 kb). ChIP-seq coverage of H3K36me3,
H3K36me2, insulator Beaf-32 and H3K27me3 in WT, dMes-
4/NSD and HypB/dSet2 KD are shown for the same region.
Light blue and dark blue stripes indicate down-regulated
genes in dMes-4/NSD and HypB/dSet2 KD, respectively.
Red stripes in the H3K27me3 track indicate genes with
increasing H3K27me3 in dMes-4/NSD or HypB/dSet2 KD. Light
red stripe indicates inactive topological-associating
domain (TAD) (see the Materials and Methods section).
**(B)** Boxplot showing compartment Eigen values from Hi-C
contacts associated with genes regulated by dMes-4/NSD
or HypB/dSet2 specifically. Active compartments are
defined by positive Eigen values, inactive compartments are
defined by negative Eigen values. Significant differences
compared with random are validated by *t* test. **(C)** Average
profiles of H3K27me3, H3K36me2, and H3K36me3 around
H3K27me3 domain borders overlapping with genes with
specific increase in dMes-4/NSD KD (left, 633 borders) and
HypB/dSet2 KD (right, 633 borders). **(D)** Intersection matrix
testing the enrichment of genes depending on their
specific increase of H3K27me3 by either dMes-4/NSD or
HypB/dSet2 depletion (computed as normalized differential
score comparing H3K27me3 between dMes-4/NSD and
HypB/dSet2), with the presence or absence of insulators on
their promoter. Exact fisher test *P*-value shown.
**(E)** Aggregated Hi-C signal between genes showing an
increase of H3K27me3 in either dMes-4/NSD KD (left panel)
or HypB/dSet2 KD (right panel) under WT condition. Median
of observed contact frequencies is computed on 11,685
pairs of coordinates containing dMes-4/NSD-regulated
genes and 12,065 pairs of coordinates containing HypB/
dSet2-regulated genes on ±20 kbp around anchor
coordinates. **(F)** Plots showing the densities of bins
harboring increasing H3K27me3 levels depending on the co-
localization (±2 kb) of inactive TAD boundaries with
H3K27me3 borders (left, 620 TAD boundaries) or not (right,
668 TAD boundaries, as defined in Fig S5D) in dMes-4/NSD
and HypB/dSet2 KD. Density of bins is shown, respectively,
to distances with inactive TAD boundaries (position 0,
oriented with the inactive TAD systematically on the left). The
same group of random bins (in grey) was chosen as a
control for both types of borders and for analysis upon
HypB/dSet2 KD or dMes-4/NSD depletions (see the Materials
and Methods section). Complementary density plots are
shown in Fig S5E. **(G)** Boxplot showing H3K27me3 ChIP-seq
reads quantifications upstream or downstream of TAD
boundary with a domain borders or not, in dMes-4/NSD
(left) or HypB/dSet2 (right) KD compare with WT condition.
Significant differences compared with control condition are
validated by Wilcoxon test.

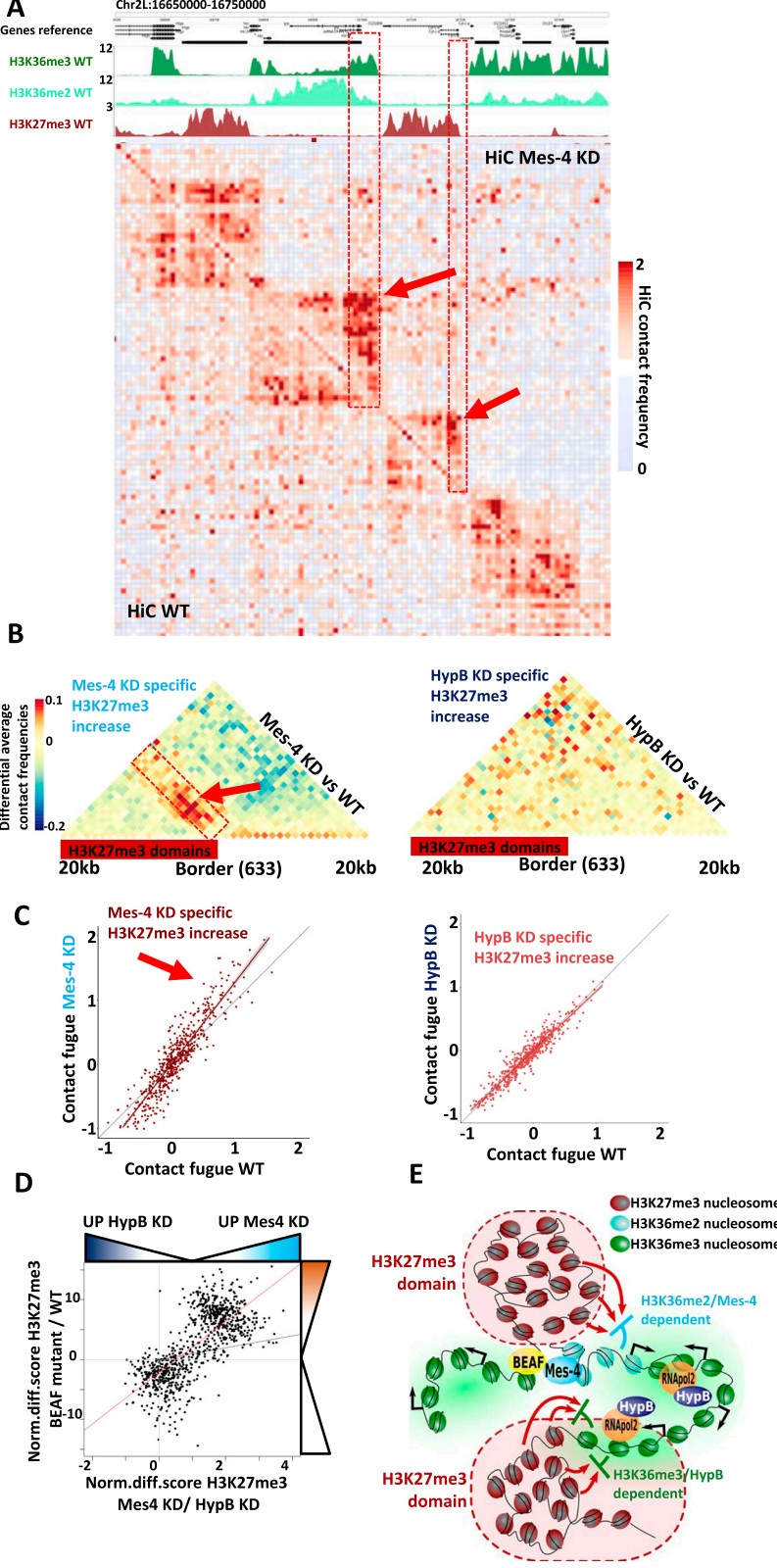

**Figure 6. dMes-4/NSD, but not HypB/dSet2, protects genes from H3K27me3 silencing in combination with 3D contact restriction at domain borders.**

**(A)** Hi-C contact matrix showing increase of contact at the borders of H3K27me3 domains in dMes-4/NSD KD compared with WT condition. **(B)** Aggregation plot of Hi-C contact at the border of H3K27me3 domain showing average differences in contact between dMes-4/NSD KD and WT or HypB/dSet2 KD and WT for 633 borders with H3K27me3 increase in dMes-4/NSD KD (left) and 633 borders with H3K27me3 increase in HypB/dSet2 KD (right). Complementary plot in Fig S6G and H. **(B, C)** Scatter plot showing quantification of contact signal over the area defined in the red rectangle in panel (B) for 633 borders with H3K27me3 increase in dMes-4/NSD KD (left) and 633 borders with H3K27me3 increase in HypB/dSet2 KD (right), comparing WT and KD signal. Complementary plot in Fig S6I and J. See Fig S6F for contact fugue score. **(D)** Scatter plot showing the correlation between the increase in H3K27me3 levels observed in dMes-4/NSD KD relative to HypB/dSet2 KD with the increase in H3K27me3 observed in insulator-looping mutants (protein Beaf-32) compared with control (Heurteau et al, 2020). The red line indicates the regression line. Pearson test (correlation = 0.72, $P$-value = $5 \times 10^{-139}$). **(E)** Model to explain how genes are protected from H3K27me3 silencing by topology-coupled mechanism involving dMes-4/NSD and H3K36me2 or transcription-coupled mechanism involving Hypb/dSet2 and H3K36me3. dMes-4/NSD protects genes in active compartments, involving insulator proteins and 3D topology of the genome, whereas HypB/dSet2 protects genes exposed to high H3K27me3 levels inside inactive compartments, by maintaining H3K36me3 to protect gene bodies.

increase was observed on the total population of such sites specifically in dMes-4/NSD depletion (Fig 6B; see red arrow; see also Fig S6A–H), unlike that of Hypb/dSet2 (Figs 6B and S6B and C; compare left and right aggregated matrices). The increase in 3D contacts occurs aside the repressive H3K27me3 domains, in absence of significant changes in contacts with more distant sites or global changes in compartmentalization (Fig S6D and E). Moreover, such detected increase in Hi-C contacts was reproducibly quantified at levels of single loci solely in dMes-4/NSD depleted cells compared with wild-type cells, (Figs 6C and S6G; see red arrow), which was not detected for control sites including those affected by Hypb/dSet2 depletion (Figs 6C and S6H–J).

Our above analyses highlight a role of dMes-4/NSD in maintaining genes flanked by insulators and TAD borders away from nearby repressive environment. Of note, mutants of Drosophila insulator proteins that specifically impaired 3D looping similarly led to H3K27me3 spreading over heterochromatin borders, depending on levels of insulator-mediated LRIs (Heurteau et al, 2020). We thus sought to compare the influence of insulator mutants on H3K27me3 levels, with that of dMes-4/NSD or Hypb/dSet2. Strikingly, unlike Hypb/dSet2, dMes-4/NSD co-regulated H3K27me3 levels over the same sites also protected from spreading by insulators, confirming that these two KMTs protect two separated gene populations as highlighted by scatter plots (Fig 6D), and as validated statistically using Fisher exact tests (Fig S6K). Therefore, chromatin hubs appear to reinforce the influence of dMes-4/NSD in protecting genes from H3K27me3, whereas Hypb/dSet2 may act independently of such 3D organization, inside heterochromatin contexts, further validating the distinction between two populations of H3K27me3 borders.

Taken altogether, our data thus unravel that although dMes-4/NSD and Hypb/dSet2 are involved in highly related H3K36 methylated states, these KMTs could preferentially antagonize H3K27me3 silencing depending on distinct contexts, thereby reinforcing TAD-based demarcation or transcription-coupled H3K36me epigenetically (Fig 6E).

## Discussion

Our data highlight two possible mechanisms for how KMTs of H3K36 may protect genes from H3K27me3 repression. In the first model, the demarcation from heterochromatin may involve H3K36me3 in antagonizing H3K27me3 spreading over genes localized within repressive heterochromatin domains. This occurs through transcription-coupled H3K36 trimethylation by Hypb/dSet2, hence requiring genes to block H3K27me3, in agreement with the interaction of Hypb/dSet2 with Pol II to mediate H3K36me3 along transcription. Such mechanism may protect moderately expressed genes particularly prone to H3K27me3 spreading, that is, as for genes within a repressive TAD. The second mechanism further involves chromatin domains in 3D. In this instance, impairing H3K36me2 appears to be sufficient to spread repressive TADs to the nearby genes, across topological borders. The spreading of H3K27me3 may then be accompanied by the extension of the topological domain. This may account, at least in part, for how epigenetic factors may contribute to topological

demarcation and the folding of chromatin into TADs in Drosophila (Jost et al, 2014) and of evolutionary conserved, transcription-coupled compartmentalized domains (Rowley et al, 2017). Of note, the depletion is induced by prolonged exposure to dsRNAs and we cannot exclude that some genes may be indirectly down-regulated thereby leading to increasing H3K27me3 levels. This scenario is however not a general property of the down-regulated genes, and neither a condition to observe H3K27me3 spreading as shown (Fig S4G). Remarkably, our data are in line with recent work in human cells, highlighting a role of the dMes-4/NSD human homolog, hNSD2, in defining active chromatin hubs in 3D, which was involved in oncogenic transcriptional programs (Lhoumaud et al, 2019) though newly formed H3K36me domains in human might not involve boundary factors as in Drosophila (Lhoumaud et al, 2014). Accordingly, altering CTCF insulators may not globally alter H3K27me3 levels at repressive domains (Schwartz et al, 2012), and additional cofactors such as MAZ may be needed as shown in mammals (Ortabozkoyun et al, 2022).

H3K36 methylation has been directly involved in regulating H3K27 methylation from C. elegans to humans by directly impeding on enzymatic activity of PRC2 (Gaydos et al, 2012). H3K36me3 is the mark that counteracts H3K27me3 for the genes exposed to high H3K27me3 levels, inside inactive TADs. In this instance, absence of insulators and TAD borders may exacerbate the need for transcription-coupled deposition of H3K36me3 by HypB/dSet2. H3K36me3 involves the loading of HypB/dSet2 onto Pol II phosphorylated by Cdk9/PTEF-b that induces transcriptional elongation (Li et al, 2003; Albert et al, 2014). Accordingly, blocking transcription per se has some influence on H3K27me3 levels of moderately expressed genes. Maintenance of H3K36me3 may thus allow a memory compatible with the timescales of Pol II transcription, by preventing H3K27me3 to invade gene bodies between two rounds of transcription cycle, thereby sustaining the expression of moderately expressed genes.

Chromatin impedes on transcriptional programming depending on the regulation of promoter accessibility by chromatin remodelers (Kornberg & Lorch, 1999). H3K36 methylation may also serve to recruit PWWP-domain Isw1 and CHD1 chromatin remodelers that regulate histone exchange and nucleosome positioning along gene bodies (Smolle et al, 2012; Venkatesh & Workman, 2013; Lhoumaud et al, 2014), which may impede on cryptic or antisense transcription (Carrozza et al, 2005; Venkatesh et al, 2016; Neri et al, 2017). PWWP-mediated recruitment of remodelers supports a major role for H3K36 methylation as key histone post-translational modifications marking euchromatin domains. In C. elegans, MES4 mediates both di- and tri-methylation thereby contributing to inheritance of H3K36-methylated active states through multiple cell divisions. MET-1 induces transcription-coupled methylation of H3K36 yet with no impact on transgenerational inheritance (Kreher et al, 2018). A role of H3K36 methylation in inheritance further relies on turnover dynamics depending on KDM4 H3K36 demethylases (Lin et al, 2012) that were shown to interact with heterochromatin protein 1 in Drosophila (Lin et al, 2008). On the one hand, removal of demethylases can condition transmission of repressive methylated histone marks (Audergon et al, 2015; Ragunathan et al, 2015). On the other hand, compartmentalization in 3D may contribute to the maintenance of H3K36 methylation state by buffering turnover

dynamics (Cuvier & Fierz, 2017). Additional factors interacting with H3K36me, such as the worm homolog of MRG15 (MRG-1), may help maintaining active compartments sequestered with MRG-1 and CBP/P300, away from silenced compartments at the nuclear periphery (Cabianca et al, 2019). Given the interplay of H3K27 and H3K36 methylation with co-transcriptional histone (de)acetylation along with chromatin remodelers (Venkatesh & Workman, 2013; Lhoumaud et al, 2014), it will be interesting to integrate how other transcription-coupled chromatin transactions including nucleosome sliding, eviction or positioning, enact onto the hidden dynamics of "spreading" of H3K27me3.

The HMT dMes-4/NSD protects genes from H3K27me3 spreading can be recapitulated by insulator looping mutants that impair chromatin looping. H3K27me3 spreading may thus depend on two types of long-range interactions. Those within inactive TADs involve self-assembly properties of PRC2 components for PRE-based H3K27me3 establishment and spreading in 1D and 3D (Oksuz et al, 2018). Those associated with transcription-coupled loop extrusion, at the periphery of such silent domain, which may restrain repressive TADs. The interactions marking repressive TADs spread beyond their borders when compromising dMes-4/NSD. Therefore, dMes-4/NSD counteracts H3K27me3 spreading by contributing to prevent interactions with active domains. This occurs in absence of changes in compartmentalization (Fig S6D), in line with recent findings showing how TADs may be regulated independently from compartments (Zenk et al, 2021). The role of insulator-binding proteins in recruiting dMes-4/NSD (Lhoumaud et al, 2014), may possibly participate to the establishment of nucleation sites for persistent and long-term memory euchromatin domains (Erdel, 2017). Of interest, a similar action has been revealed for the human 4; 14 translocation causing myeloma upon NSD2 overexpression, which is also associated with H3K36me2-insulated domains shrinking H3K27me3 involving 3D organization (Lhoumaud et al, 2019). At molecular scales, our results are consistent with biophysics and computational models in which cross talks between nucleosomes enable a memory of epigenetic state (Erdel, 2017; Alabert et al, 2020). Mainly adapted from heterochromatin domain dynamics, they are applicable to euchromatin contexts: extended H3K36me2 domains may contribute to separate euchromatin islands from surrounding heterochromatin; extended H3K36me3 over gene bodies may further exert its antagonism against H3K27me3, depending on Pol II dynamics.

Active transcription per se is thought to play a key role in chromatin compartmentalization in 1D or 3D (Berry et al, 2017), as supported by monitoring H3K27me3 in vivo (Hosogane et al, 2016). The present work suggests that an indirect contribution may actually rely on the coupling of transcriptional elongation with H3K36me3 that antagonizes H3K27me3 in 1D, whereas H3K36me2 would account for further action involving 3D compartmentalization and insulators. H3K27me3 domains further rely on mechanisms assembling heterochromatin domains involving positive feedback loop between H3K27me3 recruiting PRC2 and the HMT activity of the EED subunit of this complex (Margueron et al, 2009) or more directly Suz12-mediated PRC2 loading (Højfeldt et al, 2018). Furthermore spreading of H3K27me3 in 3D may be regulated depending on more subtle changes of chromatin states hiding additional chromatin dynamics. Of note, hundreds of genes exposed to spreading have their

transcription start sites localized at euchromatin–heterochromatin borders whereas their bodies localize in heterochromatin. Furthermore dynamics involve the ability of H3K36me to recruit DNMT3A for furthermore DNA methylation, as shown in mouse cells (Weinberg et al, 2019). H3K36me may also recruit additional PWWP-containing factors, such as the PHF19 subunit of PRC2 to induce silencing (Abed & Jones, 2012), possibly accounting for further up-regulations of genes inside euchromatin. In mammals, PRC2 first assembles into nucleation sites before it further spreads into 3D (Oksuz et al, 2018). Such spreading in 3D may also occur in *Drosophila* involving insulator-mediated long-range contacts (Heurteau et al, 2020). It may thus be interesting to test whether H3K36 methylation can regulate PRC2 nucleation sites, or if it only influences subsequent spreading in 3D. In turn, multiple players including H3K36 methylation may likely contribute to regulate compartmental domains along with transcriptional activity (Rowley et al, 2017).

# Materials and Methods

### Cell culture, RNAi, treatment, and gene expression analyses

Exponentially growing S2 cells were depleted by dsRNAs against dMes-4/NSD or Hypb/Set2 compared with mock-depletions (dsRNAs against luciferase) as previously described (Lhoumaud et al, 2014; Liang et al, 2014). Preparation of dsRNAs was done using the indicated oligos by T7-driven transcription (Fermentas TranscriptAidTM T7 High Yield Transcription Kit). Depletions were verified by quantitative RT-qPCR analysis using cDNAs prepared from control, dMes-4/NSD-, or Hypb/Set2-depleted cells, with the indicated oligos. Gene expression analyses by RNAseq were performed as previously described (Lhoumaud et al, 2014; Liang et al, 2014) in cells depleted of dMes-4/NSD or Hypb/Set2 compared with control (GSE146992). For analysis of Pol II pausing, cells were treated with either flavopiridol (3055; Sigma-Aldrich) at 1 $\mu$M during 30 min, or DRB (D1916; Sigma-Aldrich) at 50 $\mu$M during 30 min, or DMSO control (23500-260; VWR).

### Chromatin immunoprecipitation analyses

10 millions of Schneider S2 cells were cross-linked with 0.8% formaldehyde (FA) (F1635; Sigma-Aldrich) for 10 min. Crosslinking was stopped with 150 mM glycine. After two washes with PBS1X NaBu 10 mM, cells were permeabilized for 20 min with 500 $\mu$l PBS1X 0.2% Triton X-100 and 10 mM NaBu. After centrifugation, pellets were washed with lysis buffer (LB: NaCl 140 mM, HEPES, pH 7, 6 15 mM, EDTA, pH 8, 1 mM, EGTA 0.5 mM, Triton X-100 1%, sodium deoxycholate 0.1%, DTT 0.5 mM, sodium butyrate 10 mM, protease inhibitor 1X [04693124001; Roche]) and resuspended in LB + 1% SDS and 0.5% N-lauroylsarcosyl for 30 min at 4°C. Samples were sonicated for 5 × 30 s on–off cycles at 4°C (Bioruptor Pico; Diagenode). Protein A or protein G beads were coated with 0.1 mg/ml BSA NEB for 2 h at 4°C. 5 $\mu$g of antibodies were mixed with 20 $\mu$l beads and incubated overnight at 4°C (in LB 0.1% SDS). Meanwhile, chromatin samples were pre-cleared overnight at 4°C in LB with 10 $\mu$l of beads. After four washes with LB + 0.1% SDS, antibody-coupled beads were

incubated 4 h with pre-cleared chromatin. After four washes with LB + 0.1% SDS, two washes with TE 1X, elution was carried out at 70°C for 20 min in 10 mM EDTA, 1% SDS, and 50 mM of Tris–HCl (pH 8.0). Crosslinking was reversed overnight at 65°C. IP samples and inputs were incubated at 37°C for 30 min with RNAse A, and then at 55°C for 2 h with 250 $\mu$l of TE + 140 mg/ml of glycogen and 400 $\mu$g/ml of proteinase K. DNA was extracted by phenol–chloroform followed by incubation with 1.3 ml of 100% ethanol for 30 min at –80°C and centrifugation (30 min, 14,000$g$ at 4°C). DNA pellets were washed twice (70% ethanol), dried, and resuspended in $H_2O$. Chromatin immunoprecipitations were quantified by qPCR (ChIP-qPCR, primers list available in Table S2) or high-throughput sequencing (ChIP-seq). Data are available at GSE146993. Sources of commercially available antibodies were as follows: H3K36me1 (AB9048; Abcam), H3K36me2 (AB9049; Abcam for ChIP-qPCR and WB, #07-369; Upstate for ChIP-seq), H3K36me3 (AB9050; Abcam), H3K27me3 (#07-449; Upstate), Pol II total (MA1-26249 monoclonal 8WG16; Thermo Fisher Scientific).

### Bioinformatic analyses of high-throughput sequencing data

Scripts related to data processing, data integration, visualization, and statistical analysis are available on our GitHub: https://github.com/CuvierLab/depierre_perrois_H3K27K36_analysis_scripts.

### ChIP-seq data processing

Adapter sequences were trimmed (cutadapt 1.8.3 [Martin, 2011]) and checked for sequencing quality (FastQC v0.11.7 [Andrew, 2010]) using Trim Galore tool (Trim Galore 0.4.0 [Krueger, 2015]) before aligning single-stranded ChIP-seq reads (H3K27me3, H3K36me2, H3K36me3, and Pol II) on *D. melanogaster* reference genome r6.13/dm6 (http://ftp.flybase.net/genomes/Drosophila_melanogaster/dmel_r6.13_FB2016_05/) using Burrows-Wheeler Aligner (BWA 0.7.15 [Li & Durbin, 2009]). Uniquely aligned reads were filtered using Samtools 1.3.1 (Li et al, 2009) using "XT:A:U" tag in SAM files. Coverage files of aligned reads were obtained using bamCoverage tool from deepTools (deepTools 2.5.3 [Ramírez et al, 2014]) with a bin size of 10p and normalized with RPGC. Quantification of reads was performed using the computeMatrix command from deepTools using "scale-regions" over the annotated genes (UCSC dm6 downloaded from Bioconductor (https://bioconductor.org/packages/release/data/annotation/html/TxDb.Dmelanogaster.UCSC.dm6.ensGene.html).

### H3K27me3 domain calling and differential analyses

H3K27me3 domains were detected using normR R packages (Helmuth et al, 2016 *Preprint*) with a bin size configuration of 200 bp that were filtered according to their significant enrichment in H3K27me3 (compared with input) with an FDR of $1 \times 10^{-4}$. Detected H3K27me3 domains were then merged when separated by non-mappable regions to avoid considering non-mappable regions as H3K27me3 domain borders (see GitHub). The same minimal size of >1,500 bp was chosen for H3K27me3 heterochromatin domains or the intervening euchromatic domain. Differential analysis of H3K27me3 variation (e.g., Fig 4D) was performed using the normR package to compare dMes-4/NSD or HypB/dSet2 knockdown

("KD") condition compared with WT ("WT control") condition using a bin size configuration of 2,000 bp and an FDR of $1 \times 10^{-3}$. For gene-based and border-based analysis, the normalized differential score (referred as norm. diff. score) was computed as follows:

$$\text{norm. diff. score} = \frac{(KD - WT)}{\sqrt{(KD + WT)/2}}.$$

(i.e., difference between KD and WT weighted by square root of mean signal).

### RNA-seq data processing

After trimming adapters (cutadapt 1.8.3 [Martin, 2011]) and a quality control step (FastQC v0.11.7 [Andrew, 2010]) using Trim Galore tool (Trim Galore 0.4.0 [Krueger, 2015]), RNA-seq reads in wild-type, dMes-4/NSD KD, and HypB/dSet2 KD were aligned on *D. melanogaster* reference genome r6.13/dm6 (http://ftp.flybase.net/genomes/Drosophila_melanogaster/dmel_r6.13_FB2016_05/) using STAR Aligner (STAR 2.5.2b [Dobin et al, 2013]). Processing of aligned reads to filter and sort them was performed using Samtools 1.3.1 (Li et al, 2009). Coverage files of aligned reads was obtained using bamCoverage tool from deepTools (deepTools 2.5.3 [Ramírez et al, 2014]) with a bin size of 10p and normalized over genomic content by RPGC. Quantification of reads on gene bodies was computed with computeMatrix command from deepTools using the "scale-regions" option of genes annotated from UCSC dm6 (downloaded from Bioconductor).

### Differential expression analysis

RNA-seq reads were filtered using HTSFilter followed by differential expression analysis performed on RNA-seq replicates with the R package DESeq2 (Love et al, 2014). Significant changes in expression log fold change were scored and selected providing a student test $P$-value < 0.05. Clustering of co-regulated genes (shown in Fig 5C) was carried out using the Euclidean distance of the hclust R function for log fold changes of all genes. Clustering analysis performed on genes with logFC > 0.3 resulted in seven significantly distinct clusters of co-regulated genes.

### Hi-C data generation and processing

Hi-C data pertaining to this study (GEO: GSE146994) were generated using the genome-wide Hi-C kit from Arima Genomics S2 cells treated with siMes-4/NSD and siHypB/dSet2 compared with control siLuc (control luciferase gene that does not exist in Drosophila). Hi-C data were processed using a standard pipeline using Juicer (Durand et al, 2016). Adapter sequences were trimmed (Trim Galore version 0.4.0; cutadapt 1.8.3) and reads were then indexed and aligned with bwa-0.7.15 followed by reads filtering (samtools-1.3.1; bash) and conversion to bigWig format to produce a quantification matrix (Python-3.4.3/deepTools-2.5.3). The obtained filtered Hi-C contact matrices (see Table S3 and Hi-C files in GSE146994) were visualized using 2D plots (e.g., Fig 5A) using Juicebox (https://www.aidenlab.org/juicebox/) either as observed (upper row) or

observed/expected (lower row), at 1 kb resolution, using the sqrt format coverage (sqrt). The same procedure was applied to our data or to external data (Ramírez et al, 2018), showing similar distributions of contacts (Fig 5A). Hi-C–processing statistics are provided in Table S3. Analyses of Hi-C data from HypB/dSet2 KD and dMes-4/NSD KD was performed compared with our control Hi-C data (in control cells) performed in parallel. Compartment calling defining compartment Eigen values (Figs 6 and S6) was carried out using Juicer tool Eigen vector. Hi-C data used in Fig 5 were downloaded from the study of Ramírez et al (2018). Aggregated or averaged plots of Hi-C contact signal are performed following the method previously used as described in the studies of Liang et al (2014); Durand et al (2016); Ing-Simmons & Vaquerizas (2019); Heurteau et al (2020), which consists in extracting and averaging sub-matrices corresponding to the projections of genomic loci of interest (transcription start sites, Beaf-32–binding sites, or H3K27me3 domain borders).

### Visualization and statistical analysis

All analysis and visualization were performed with R 3.4.3 version using ggplot2 (Wickham, 2009) adapted functions. *P*-values were obtained by Wilcoxon or Fisher exact tests. Profiles were plotted using SeqPlots R package from Bioconductor (SeqPlots 1.16.0 [Stempor & Ahringer, 2016]). For Fisher exact tests, genes were systematically ranked according to the indicated feature and split into groups of the same number of genes. Gene set enrichment analysis was performed using fgsea Bioconductor R package (Subramanian et al, 2005; Korotkevich et al, 2019 *Preprint*). Dendrograms (Fig 3; EV3) were performed after center, normalizing all data and by taking projections of each feature onto the first three major PCs obtained from principal component analysis. The result is shown as a dendrogram, showing the relative distances (y-axis; 1/correlation) among all features (i.e., the shorter the distance/height between two features the more they are correlated).

# Data Availability

All RNA-seq, ChIP-seq, and Hi-C data pertaining to this study were deposited to GEO of NCBI (GSE146994). External data were downloaded from NCBI GEO data with following accession numbers: H3K9me3 from GSE99027 (Colmenares et al, 2017); H3 total from GSE113470 (Tettey et al, 2019); H3K27ac, H3K4me1, and H3K4me3 from GSE85191 (Henriques et al, 2018); H3K27me3 WT and mutant BEAF from GSE130211 (Heurteau et al, 2020); external H3K27me3 from GSM2776906 (Huang et al, 2017); H3K36me3 WT from GSM2776903 (Huang et al, 2017); H3K36me2 WT from GSM3106537 (Tettey et al, 2019). Hi-C data were downloaded from GSE97965 (Ramírez et al, 2018).

# Supplementary Information

# Acknowledgements

We thank Fabian Erdel for critical reading of the article and Julien Anglade for help with ChIP-seq. We also thank the BGI and the Genotoul platform for technical facility. The laboratory of O Cuvier was supported by a "FRM team" grant from the Fondation Recherche Médicale (grant number DEQ20160334940) and by ANR (HELICO). The group was also supported by fellowships of the Ministry of Research and Technology (MRT; O Fosseprez and D Depierre), La Ligue Nationale contre le Cancer (LNCC), the CNRS and the Inserm (O Cuvier).

## Author Contributions

D Depierre: software, investigation, and writing—review and editing.
C Perrois: formal analysis, validation, investigation, and writing—review and editing.
N Schickele: software, validation, investigation, and writing—review and editing.
P Lhoumaud, A Heurteau, O Fosseprez: formal analysis, validation, and investigation.
M Abdi-Galab: investigation.
R Margueron: conceptualization and writing—review and editing.
O Cuvier: conceptualization, resources, formal analysis, supervision, funding acquisition, investigation, methodology, writing—original draft, and project administration.

## Conflict of Interest Statement

The authors declare that they have no conflict of interest.

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
