## [Reviewer comments · Life Science Alliance]

Life Science Alliance

Chromatin in 3D distinguishes dMes-4/NSD and Hypb/dSet2 in protecting genes from H3K27me3 silencing

David Depierre, Charlene Perrois, Naomi Schickele, Priscillia Lhoumaud, Mahdia Abdi-Galab, Olivier Fosseprez, Alexandre Heurteau, Raphael Margueron, and Olivier Cuvier

DOI: <https://doi.org/10.26508/lsa.202302038>

Corresponding author(s): Olivier Cuvier, Center for Integrative Biology (CBI/ UMR5077)-CNRS - Universite of Toulouse

Review Timeline:

Submission Date:	2023-03-14
Editorial Decision:	2023-03-15
Revision Received:	2023-07-24
Editorial Decision:	2023-07-26
Revision Received:	2023-08-25
Accepted:	2023-08-28

Transaction Report:

Please note that the manuscript was previously reviewed at another journal and the reports were taken into account in the decision-making process at *Life Science Alliance*. Since the original reviews are not subject to Life Science Alliance's transparent review process policy, the reports and author response cannot be published.

March 15, 2023

Re: Life Science Alliance manuscript #LSA-2023-02038-T

Olivier Cuvier
Center for Integrative Biology (CBI-CNRS/University of Toulouse)

Dear Dr. Cuvier,

Thank you for submitting your manuscript entitled "Chromatin organization in 3D distinguishes the influences of dMes-4/NSD and Hypb/dSet2 in protecting genes from H3K27me3 silencing" to Life Science Alliance. We invite you to submit a revised manuscript addressing the Reviewer points.

Thank you for this interesting contribution to Life Science Alliance. We are looking forward to receiving your revised manuscript.

Sincerely,

B. MANUSCRIPT ORGANIZATION AND FORMATTING:

July 26, 2023

RE: Life Science Alliance Manuscript #LSA-2023-02038-TR

Dr. Olivier Cuvier
Center for Integrative Biology (CBI/ UMR5077)-CNRS - Universite of Toulouse
Molecular Cellular and Development Biology unit (UMR5077), Laboratory of Chromatin Dynamics
rue Grunberg-Manago
Toulouse 31062
France

Dear Dr. Cuvier,

Thank you for submitting your revised manuscript entitled "Chromatin in 3D distinguishes dMes-4/NSD and Hypb/dSet2 in protecting genes from H3K27me3 silencing". We would be happy to publish your paper in Life Science Alliance pending final revisions necessary to meet our formatting guidelines.

- please make sure the author order in your manuscript and our system match
- please consult our manuscript preparation guidelines <https://www.life-science-alliance.org/manuscript-prep> and make sure your manuscript sections are in the correct order
- please add a conflict of interest statement to your main manuscript text
- please upload all figure files as individual ones, including the supplementary figure files
- please add a callout for Fig 2E, Fig S6L, Fig S6M to your main manuscript text
- please add sizes next to the blots in Figure S2

A. FINAL FILES:

B. MANUSCRIPT ORGANIZATION AND FORMATTING:

Sincerely,

August 28, 2023

RE: Life Science Alliance Manuscript #LSA-2023-02038-TRR

Dr. Olivier Cuvier
Center for Integrative Biology (CBI/ UMR5077)-CNRS - Universite of Toulouse
Molecular Cellular and Development Biology unit (UMR5077), Laboratory of Chromatin Dynamics
rue Grunberg-Manago
Toulouse 31062
France

Dear Dr. Cuvier,

Thank you for submitting your Research Article entitled "Chromatin in 3D distinguishes dMes-4/NSD and Hypb/dSet2 in protecting genes from H3K27me3 silencing". It is a pleasure to let you know that your manuscript is now accepted for publication in Life Science Alliance. Congratulations on this interesting work.

DISTRIBUTION OF MATERIALS:

Again, congratulations on a very nice paper. I hope you found the review process to be constructive and are pleased with how the manuscript was handled editorially. We look forward to future exciting submissions from your lab.

Sincerely,
